# Severity Estimation for Interturn Short-Circuit and Demagnetization Faults through Self-Attention Network

**DOI:** 10.3390/s22124639

**Published:** 2022-06-20

**Authors:** Hojin Lee, Hyeyun Jeong, Seongyun Kim, Sang Woo Kim

**Affiliations:** Department of Electrical Engineering, Pohang University of Science and Technology, 77 Cheongam-Ro, Nam-Gu, Pohang 37673, Korea; suvvus@postech.edu (H.L.); jhy90@postech.edu (H.J.); ksy3dmbe3kor@postech.edu (S.K.)

**Keywords:** deep learning, demagnetization fault, fault diagnosis, interturn short-circuit fault, permanent-magnet synchronous machine, self-attention, severity estimation

## Abstract

This study presents a novel interturn short-circuit fault (ISCF) and demagnetization fault (DF) diagnosis strategy based on a self-attention-based severity estimation network (SASEN). We analyze the effects of the ISCF and DF in a permanent-magnet synchronous machine and select appropriate inputs for estimating the fault severities, i.e., a positive-sequence voltage and current and negative-sequence voltage and current. The chosen inputs are fed into the SASEN to estimate fault indicators for quantifying the fault severities of the ISCF and DF. The SASEN comprises an encoder and decoder based on a self-attention module. The self-attention mechanism enhances the high-dimensional feature extraction and regression ability of the network by concentrating on specific sequence representations, thereby supporting the estimation of the fault severities. The proposed strategy can diagnose a hybrid fault in which the ISCF and DF occur simultaneously and does not require the exact model and parameters essential for the existing method for estimating the fault severity. The effectiveness and feasibility of the proposed fault diagnosis strategy are demonstrated through experimental results based on various fault cases and load torque conditions.

## 1. Introduction

Condition monitoring and fault diagnosis are fundamental processes for maintaining the advantages of permanent-magnet synchronous machines (PMSMs) for various applications. Accurate and preemptive fault diagnoses can reduce economic losses and improve reliability and stability by preventing excessive system downtime and accidents [1]. Accordingly, many studies have been conducted on diagnosing interturn short-circuit faults (ISCFs), demagnetization faults (DFs), bearing faults (BFs), and eccentricity faults (EFs), all of which frequently occur in PMSMs [2]. Among these, ISCFs and DFs directly reduce the efficiency of the PMSM and increase its operational cost. In addition, owing to their characteristics, each of these faults can lead to the other and/or increase the other’s severity [3]. Therefore, it is essential to accurately diagnose ISCFs and DFs at an early stage.

The ISCF is one of the most frequently occurring stator winding failures, and results in a short circuit owing to a breakdown of the insulation between adjacent windings [4]. The main causes of ISCFs are mechanical, electrical, and thermal stresses [5]. A large amount of circulating current is generated in the short-circuited winding; this increases the torque ripple owing to the phase imbalance, reducing the efficiency and performance of the PMSM, and endangering its safe operation [6]. In addition, excessive local heat is generated, accelerating the insulation breakdown and further increasing the severity of the ISCF [7]. Furthermore, the permanent magnets (PMs) in a rotor can be irreversibly demagnetized, owing to the reductions in the magnetic coercivity and local inverse magnetic field according to the large circulating current [8]. Therefore, it is essential to diagnose an ISCF at an early stage before it becomes serious and leads to other failures, such as DFs.

A DF indicates that the strength of the permanent magnet in the PMSM has been irreversibly reduced. The main causes of DFs are temperature rises owing to operation, armature reactions, reverse magnetic fields owing to stator currents, and aging [9]. A DF directly reduces the performance and efficiency of the PMSM, as the electromagnetic torque of the PMSM is proportional to the cross-product between the current vector and PM flux linkage vector [10]. When a DF occurs in a PMSM, a larger input current is required to maintain the same output torque, leading to increased copper loss and higher heat. This can cause more severe irreversible demagnetization, and the high heat can damage the insulation of the stator windings [3]. Therefore, it is important to diagnose a DF before it worsens to avoid serious damage to the PMSM and increase the reliability of the application system.

Many researchers have studied approaches to diagnosing ISCFs and DFs. For example, motor current signature analysis (MCSA) uses the fast Fourier transform, continuous wavelet transform, and Hilbert–Huang transform to detect the faults by analyzing the harmonics of the phase current caused by the faults [11,12,13]. However, MCSA does not accurately estimate the severity of failure, and there is a risk of false detection due to other failures such as the EF [14].

A model-based fault diagnosis method for estimating the severity of a fault has also been proposed. Methods have been proposed for estimating fault indicators by using zero-sequence components and negative-sequence components, with the aim of diagnosing ISCFs [15,16]. In [17], a method was studied for diagnosing ISCFs based on residual current by using a dynamic model. Parameter estimation methods based on a mathematical model of the PMSM have also been proposed for diagnosing DFs [18]. However, the accuracy of model-based fault-diagnosis methods is highly dependent on the accuracy of the model. Changes in model parameters owing to the occurrence of other faults directly influence the reliability and robustness of the fault diagnosis, as the existing methods assume that only one fault occurs. For example, in the method in [17], if a DF occurs, an ISCF may be misdiagnosed, because the residual current vector changes owing to the decrease in the amplitude of the PM flux from the DF. In addition, ISCFs and DFs can induce each other. As such, a method for diagnosing both faults is essential, but has not yet been studied. Therefore, it is necessary to develop a method for accurately diagnosing both ISCFs and DFs.

In view of recent developments in deep learning, data-driven fault-diagnosis methods have been widely studied. In [19], a detection method for DFs and BFs in a PMSM was studied; it used a Visual Geometry Group-16 network with signal-to-image conversion. Similarly, a 1-D convolutional neural network (CNN) composed of multi-scale feature extraction modules has been proposed for detecting DFs and BFs [20]. In [21], a CNN-based diagnostic method for DFs and mixed damages using raw stator current signals was proposed. Also, a detection and classification method for the incipient ISCF using a CNN was presented [22]. In [23], a sparse autoencoder was applied to detect ISCFs in a PMSM by using data augmentation and a conditional generative adversarial network. In [24], a transformer convolution network (TCN) was proposed, with transfer learning for complementing a limited number of training samples. In [25], an attention recurrent neural network (RNN) was applied to estimate the severity of ISCFs for early-stage diagnoses by using only current signals. Data-driven fault diagnosis methods using deep learning have shown remarkable results; however, they are mainly directed to only fault detection and classification, and studies on estimating the fault severity are relatively rare. An estimation of fault severity can provide a reliable guide for establishing safe operating areas for PMSMs, and can be applied to fault-tolerant controls and maintenance schedules. Therefore, it is essential to develop a method for estimating the severity of faults, rather than simply detecting them.

This study proposes a novel fault-diagnosis strategy for estimating the severity of ISCFs and DFs in a PMSM by using a self-attention-based severity estimation network (SASEN) inspired by the transformer [26]. The SASEN consists of an encoder and decoder, both of which comprise self-attention modules. A multi-head self-attention (MSA) module, in which self-attention modules are configured in parallel, is advantageous for extracting various high-dimensional features from input data. In addition, the SASEN concentrates on certain features of the encoder through the encoder–decoder attention process, thereby improving the regression performance. The SASEN receives a positive-sequence voltage (PSV), positive-sequence current (PSC), negative-sequence voltage (NSV), and negative-sequence current (NSC) as inputs, and outputs two fault indicators directly related to the severity of the ISCF and DF. The network can diagnose ISCFs and DFs by estimating the severity of faults through regression. In addition, the proposed strategy can diagnose a hybrid fault (HF) in which the ISCF and DF occur simultaneously. This is believed to be the first approach that can diagnose the HF. The experimental results validate the effectiveness and feasibility of the proposed diagnosis strategy for estimating the severities of ISCFs, DFs, and HFs under various load torques and fault conditions.

The main contributions of this study are as follows.


The proposed SASEN can be used to evaluate the severity of ISCFs and DFs. By applying the self-attention mechanism, the SASEN provides superior model representation, feature extraction, and regression capabilities. The proposed strategy can be extended to the diagnosis of other faults.The proposed strategy can diagnose an HF, i.e., when the ISCF and DF occur simultaneously. To the best of our knowledge, this is the first study on diagnosing an HF by estimating its severity.Fault diagnosis is achieved for various load torques and fault conditions. In particular, the proposed strategy can diagnose faults even under untrained load torques. Therefore, it has an excellent generalization ability and is more effective, as it is not necessary to train all possible load torques.The proposed strategy can accurately diagnose faults without requiring the exact model and parameters necessary for severity estimation in the conventional method.


The remainder of this paper is organized as follows. In Section 2, the impacts of ISCFs and DFs on the PMSM are analyzed, aiming to select the inputs and outputs for the severity estimation network. In Section 3, a SASEN is proposed for the diagnosis of ISCFs, DFs, and even HFs. In Section 4, the effectiveness and feasibility of the proposed diagnosis strategy are demonstrated by using experimental results. Finally, Section 5 concludes the study.

## 2. Analysis of Faults

In this section, the impacts of the ISCF, DF, and the HF on the PMSM are analyzed. The changes in three-phase voltage and current due to the faults are analyzed by using the model equation of the faulty PMSM. By using the analysis, we select the inputs and outputs of the deep-learning model for estimating the severity of the faults.

### 2.1. Analysis of Interturn Short-Circuit Fault

The stator phase voltage equation for a PMSM with an ISCF in the abc reference frame is expressed as follows [3]:(1)$$\left[v_{abc}\right]=R_{s}\left[i_{abc}\right]+\frac{d}{dt}\left(\left[L_{s}\right]\left[i_{abc}\right]+\left[\psi_{m,abc}\right]\right)-R_{s}\left[P\right]\mu i_{f}-\frac{d}{dt}\left(\left[L_{s}\right]\left[P\right]\mu i_{f}\right)$$
where $\left[v_{abc}\right]={\left[\begin{array}{ccc} v_{a} & v_{b} & v_{c} \end{array}\right]}^{T}$, $\left[i_{abc}\right]={\left[\begin{array}{ccc} i_{a} & i_{b} & i_{c} \end{array}\right]}^{T}$, $\left[\psi_{m,abc}\right]=\psi_{m}\left[\begin{array}{c} \cos\left(\theta \right)\\ \cos\left(\theta -\frac{2}{3}\pi \right)\\ \cos\left(\theta +\frac{2}{3}\pi \right) \end{array}\right]$, $\left[L_{s}\right]=\left[\begin{array}{ccc} L & M & M\\ M & L & M\\ M & M & L \end{array}\right]$, and $\left[P\right]={\left[\begin{array}{ccc} 1 & 0 & 0 \end{array}\right]}^{T}$. $\left[v_{abc}\right]$ and $\left[i_{abc}\right]$ are the stator-phase voltage and current vectors, respectively. $\left[\psi_{m,abc}\right]$ is the flux linkage vector, $\psi_{m}$ is the amplitude of the PM flux and θ is the electrical angle of the rotor. $R_{s}$ is the stator resistance. [P] represents the location where the ISCF occurs. [P] becomes ${\left[\begin{array}{ccc} 0 & 1 & 0 \end{array}\right]}^{T}$ or ${\left[\begin{array}{ccc} 0 & 0 & 1 \end{array}\right]}^{T}$ when the ISCF occurs in phases *b* or *c*, respectively. $\left[L_{s}\right]$ is the stator inductance matrix, and *L* and *M* denote the self and mutual inductances, respectively. μ is the shorted turn ratio. $i_{f}$ is the fault current through the shorted circuit.

The ISCF can be modeled as an additional circuit, owing to the short between windings with broken insulation in a single phase. A large amount of circulating current flows through this circuit, significantly affecting the PMSM. The fault current depends on μ and the fault resistance $r_{f}$; as μ increases or $r_{f}$ decreases, the magnitude of the fault current increases. As shown in (Equation 1), new components are generated when an ISCF occurs. These fault components are the largest in the phase where the fault occurs, and break the balance between phases. This results in imbalances in the three-phase impedance and current. In addition, the magnitude of the three-phase current increases to provide similar performance at a given load torque [3]. The degree of the imbalance and magnitude of the three-phase current can be easily quantified by using the NSC and PSC, respectively. The NSC and PSC are calculated as follows:(2)$$i_{NS}=\frac{1}{3}\left(i_{a}+\alpha^{2}\cdot i_{b}+\alpha \cdot i_{c}\right)$$
(3)$$i_{PS}=\frac{1}{3}\left(i_{a}+\alpha \cdot i_{b}+\alpha^{2}\cdot i_{c}\right),$$
where $i_{NS}$ and $i_{PS}$ represent the NSC and PSC, respectively. α is the phase-rotation operator and is $e^{j\cdot \frac{2pi}{3}}$.

The NSC is suitable for quantifying the imbalance in the three-phase current of the PMSM. Even for a healthy PMSM, the NSC is non-zero, owing to the inherent asymmetry created by manufacturing. However, when an ISCF occurs, the NSC increases significantly compared with that of a healthy PMSM. As the severity of the ISCF increases, the imbalance of the three-phase current also increases, significantly increasing the NSC. The NSC also affects the NSV, which is calculated as follows:(4)$$v_{NS}=\frac{1}{3}\left(v_{a}+\alpha^{2}\cdot v_{b}+\alpha \cdot v_{c}\right).$$

Accordingly, the NSC and NSV can be used to estimate the severity of an ISCF. In addition, the magnitude of the three-phase current is increased to maintain the given load torque; in this context, the average magnitude of the three-phase current can be calculated by using the PSC.

### 2.2. Analysis of Demagnetization Fault and Hybrid Fault

A DF is an irreversible decrease in the flux of the PMs. In other words, the DF can be modeled as a decrease in the amplitude of the PM flux. The stator phase voltage equation of the PMSM with a DF in the abc reference frame is expressed as follows:(5)$$\left[v_{abc}\right]=R_{s}\left[i_{abc}\right]+\frac{d}{dt}\left(\left[L_{s}\right]\left[i_{abc}\right]+\left[\psi_{m,f,abc}\right]\right),$$
where $\left[\psi_{m,f,abc}\right]=\psi_{m,f}\left[\begin{array}{c} \cos\left(\theta \right)\\ \cos\left(\theta -\frac{2}{3}\pi \right)\\ \cos\left(\theta +\frac{2}{3}\pi \right) \end{array}\right]$ is the flux linkage from the demagnetized magnets, and $\psi_{m,f}$ is the amplitude of the demagnetized magnet flux. The stator phase voltage is reduced when a DF occurs, because $\psi_{m,f}$ is smaller than $\psi_{m}$. Moreover, the decrease in the magnetic flux leads to a decrease in the torque output, because the torque of the PMSM is proportional to the amplitude of the PM flux. Therefore, similar to the case with the ISCF, the magnitude of the three-phase current increases to maintain the given load torque [3]. The magnitudes of the phase voltage and current can be quantified by using the PSV and PSC. The PSC is defined above, and the PSV can be calculated as follows:(6)$$v_{PS}=\frac{1}{3}\left(v_{a}+\alpha \cdot v_{b}+\alpha^{2}\cdot v_{c}\right),$$
where $v_{PS}$ represents the PSV. The PSV provides the average magnitude of the three-phase voltage. As the DF becomes more severe, the magnitude of the stator phase voltage is further reduced, leading to a proportional reduction in the PSV. Conversely, to maintain the load torque, the PSC increases as the DF becomes more severe. Therefore, the severity of the DF can be estimated by utilizing the property that the PSV decreases and the PSC increases as the DF becomes severe.

The HF represents the simultaneous occurrence of the ISCF and DF, and has the properties of both faults. Similar to the ISCF, the HF increases the NSC, owing to a three-phase imbalance. In addition, the PSC is increased to maintain the load torque, owing to the reduced efficiency of the faulty PMSM. In addition, the PSV decreases because the flux of the PM decreases.

### 2.3. Input and Output Selection

Analyses of faults suggest that the severity of a fault can be estimated by using the PSV, PSC, NSV, and NSC, as the severity is closely related to these signals. Therefore, fault indicators for quantifying the severity of the ISCF, DF, and HF should be defined for an accurate diagnosis. Considering the factors indicating the severity of the faults, two fault indicators are defined as follows:(7)$${\mathrm{FI}}_{\mathrm{ISCF}}=\mu I_{f}$$
(8)$${\mathrm{FI}}_{\mathrm{DF}}=\frac{\psi_{m,f}}{\psi_{m,h}},$$
where $I_{f}$ represents the magnitude of the fault current, and $\psi_{m,h}$ represents the amplitude of the healthy magnet flux. ${\mathrm{FI}}_{\mathrm{ISCF}}$ and ${\mathrm{FI}}_{\mathrm{DF}}$ represent the fault indicators for the ISCF and DF, respectively. In a healthy PMSM, ${\mathrm{FI}}_{\mathrm{ISCF}}$ and ${\mathrm{FI}}_{\mathrm{DF}}$ are 0 and 1, respectively. In the case of an ISCF, ${\mathrm{FI}}_{\mathrm{ISCF}}$ increases, but ${\mathrm{FI}}_{\mathrm{DF}}$ remains 1. Conversely, in the case of a DF, ${\mathrm{FI}}_{\mathrm{DF}}$ decreases, but ${\mathrm{FI}}_{\mathrm{ISCF}}$ is zero. In the case of an HF, ${\mathrm{FI}}_{\mathrm{ISCF}}$ increases and ${\mathrm{FI}}_{\mathrm{DF}}$ decreases. Therefore, to diagnose the ISCF, DF, and HF, a nonlinear relationship is learned by using the SASEN, with the PSV, PSC, NSV, and NSC as inputs, and the fault indicators as outputs.

## 3. Proposed Severity Estimation Method

This section introduces the proposed self-attention-based severity estimation network for the diagnosis of the ISCF, DF, and the HF. First, the self-attention module underlying the SASEN is described, followed by the overall structure of the SASEN. The SASEN has an encoder–decoder structure and has excellent regression capabilities by using attention layers. Then, the overall structure of the fault diagnosis system by using the SASEN is illustrated.

### 3.1. Self-Attention Module

The attention mechanism is used to increase the performance of the deep-learning model by making the model pay attention to a specific vector. It does so by constructing a weighted combination of vectors according to specific rules. Self-attention is a type of attention mechanism, and comprises calculating a sequence expression by finding the correlations between components at different positions in a sequence. The overall structure of the self-attention module is shown in Figure 1. The scaled dot-product attention [26] is used, and is expressed as follows:(9)$$Attention\left(Q,K,V\right)=softmax\left(\frac{{\mathrm{QK}}^{T}}{\sqrt{d_{k}}}\right)V,$$
where Q, K, and V are the query, key, and value matrices, with dimensions of $d_{k}$, $d_{k}$, and $d_{v}$, respectively. $\sqrt{d_{k}}$ is a scaling factor. First, the dot products between the query and keys are calculated, and then are scaled to $1/\sqrt{d_{k}}$. The scaling factor helps to stabilize the gradient while training the network. Then, a softmax function is applied to obtain the normalized attention weights, which are multiplied by the values. Through this process, the attention module outputs a new sequence representation.

### 3.2. Self-Attention-Based Severity Estimation Network

The overall architecture of the SASEN is shown in Figure 2. Initially, input embedding is required before an input sequence can be used as an input to the network. An input sequence with dimensions of $d_{input}$ is projected as a vector with dimensions of $d_{model}$ by applying a linear projection in the input embedding. This process is essential because the dimensions of the sequence in all layers of the SASEN are constant, as $d_{model}$. In this study, $d_{model}$ is 64.

The embedded sequence is the input to the SASEN. The SASEN has an encoder–decoder structure. The encoder maps the incoming sequence to a feature representation. The encoder is composed of multiple encoder layers, each of which consists of two sublayers: an MSA, and a fully connected feed-forward network (FFN). The MSA applies the attention function to each of the *h* times linearly projected versions of the keys, values, and queries with dimensions of $d_{k}$, $d_{k}$, and $d_{v}$, respectively. That is, the MSA represents a combination of multiple self-attentions in parallel, and can be expressed as follows:(10)$$MSA\left(Q,K,V\right)=Concat\left(Head_{1},\cdots ,Head_{h}\right)W^{O}$$
(11)$$Head_{i}=Attention\left(QW_{i}^{Q},KW_{i}^{K},VW_{i}^{V}\right),$$
where the trainable linear projection matrices $W_{i}^{Q}\in R^{d_{model}\times d_{k}}$, $W_{i}^{K}\in R^{d_{model}\times d_{k}}$, $W_{i}^{V}\in R^{d_{model}\times d_{v}}$, and $W^{O}\in R^{hd_{v}\times d_{model}}$. As the MSA approach jointly uses multiple self-attention modules, it is possible to find the correlation in the sequence from various angles, thereby enhancing the expressive power of the attention layer. Therefore, the model can collect various types of information from different representation subspaces at different locations, without increasing the number of parameters.

The other sublayer, the FFN, is composed of two linear transformations with the rectified linear unit activation function in the middle. The FFN is expressed as follows:(12)$$FFN\left(x\right)=max\left(0,xW_{1}+b_{1}\right)W_{2}+b_{2}.$$

The input and output of the FFN have dimensions of $d_{model}$, and the hidden layer of the FFN has dimensions of $d_{ff}=2048$. After the FFN, the input sequences are fully expressed as feature representations. Residual connections [27] are used around the MSA and the FFN. The residual connections prevent gradient vanishing in the deep network. In addition, applying layer normalization [28] helps the training to converge stably, and shortens the training time.

The decoder receives the feature representations from the encoder and generates the target output sequences. The decoder has a similar structure as the encoder, but includes another MSA, called the encoder–decoder attention. This attention layer supports the decoding process by calculating the associations between the encoder and decoder and automatically concentrating on the relevant encoded features. Through the encoder–decoder attention, the regression performance for producing the target sequence is improved. As with the encoder, residual connections are used around the sublayers in the decoder, followed by layer normalization. Finally, the decoder outputs the target sequences. In this study, $d_{k}$, $d_{v}$, and *h* are 8, 8, and 8, respectively. The number of encoder and decoder layers (N) is two.

### 3.3. Training Procedure

In this study, the proposed SASEN model was implemented by using Python with the Pytorch deep-learning library. Based on the analysis in Section 2, the inputs of the network were four sequences, i.e., the PSV, PSC, NSV, and NSC, and the outputs were the two fault indicators ${\mathrm{FI}}_{\mathrm{ISCF}}$ and ${\mathrm{FI}}_{\mathrm{DF}}$. The learnable parameters of the SASEN were trained by standard backpropagation with an $l_{2}$ loss function, which is expressed as follows:(13)$$l_{2}\left(y,\hat{y}\right)=\frac{1}{M}\sum_{i=1}^{M}{\left({\hat{y}}_{i}-y_{i}\right)}^{2},$$
where $\hat{y}$ and y are the estimated and actual values, respectively. The diagnosis model was trained by using the Adam optimizer with mini-batches of size 100 and a learning rate of 0.0001. The total number of training epochs was 400. A dropout rate of 0.2 was applied for regularization.

### 3.4. Overall Structure of Diagnosis System

The overall structure of the fault diagnosis system is shown in Figure 3. First, the three-phase voltage and current signals measured from the PMSM were pre-processed. In particular, (Equation 2)–(Equation 4) and (Equation 6) were used to convert the three-phase voltage and current signals to the NSC, PSC, NSV, and PSV, respectively, and then were passed through a bandpass filter to reduce noise. The pre-processed signals were normalized and fed into the proposed SASEN, which outputs the fault indicators representing the severity of the ISCF and DF. At this point, the proposed network was trained to exhibit excellent feature extraction and regression capabilities by using a sufficient number of healthy and faulty samples. In general, the influences of disturbances or noise during a test can cause fluctuations in the estimated fault indicators. To compensate for this, the output of the network was post-processed by using a moving average filter, so as to provide a more reliable estimate of the fault severities.

## 4. Experimental Results

### 4.1. Experimental Setup

To demonstrate the effectiveness and feasibility of the proposed fault diagnosis strategy, experiments were conducted by using a test rig, as shown in Figure 4. The experimental setup consisted of an interior PMSM (IPMSM), dynamo machine, encoder, and data-acquisition device. The IPMSM had six poles, 36 slots, and 120 turns, with a concentrated winding in each phase. The detailed specifications of the IPMSM are listed in Table 1. The data-acquisition device acquired the stator three-phase voltage, current, and rotational speed at a sampling rate of 100 kHz. In addition, 1/20 downsampling was performed on the measured data.

The experiments were conducted for four cases, i.e., one healthy machine, and three faulty machines with the ISCF, DF, and HF, respectively. The detailed fault and operating conditions are shown in Table 2. In all of the experiments, the IPMSM was operated at a rotational speed of 4500 rpm and various load torques ranging from 1 to 4 Nm. First, to implement the ISCF in the IPMSM, we implemented a turn-to-turn short by connecting a resistor to the turns of the last winding of phase *c*, as shown in the ISCF tap in Figure 4. The resistor connected at this short circuit provided the fault resistance, and had a value of 0.218 Ω. In addition, 5, 10, and 15 turns were sequentially shorted out of a total of 120 turns to realize three different levels of fault severity. The 5-turns short was the weakest fault, and the 15-turns fault was the most serious. The fault current circulating through the short circuit was measured. When only the ISCF occurred, the magnitudes of the fault currents were 20 A and 35 A for the weakest and most severe faults, respectively.

Next, to implement the DF in the IPMSM, the PM embedded in the rotor was replaced with a dummy block with the same weight and volume but no magnetism to realize demagnetization, as shown in Figure 5. There were six permanent magnets in the IPMSM, and two levels of fault severity were implemented by sequentially replacing one or two magnets with dummy blocks. Therefore, the demagnetized IPMSM was tested with 83.3% and 66.7% residual fluxes as compared to those of a healthy machine, respectively.

The HF comprised a simultaneous implementation of an ISCF and DF. Six types of fault conditions were tested, in view of the three levels of ISCF severity and two levels of DF severity. In the HF, the characteristics of the ISCF and DF appeared simultaneously.

### 4.2. Training and Test

In total, 29,896 data samples were acquired through the experiments with healthy and faulty IPMSMs, consisting of 4067 healthy samples, 6951 ISCF samples, 5072 DF samples, and 13,806 HF samples. The sequence length of each data sample was 136. This dataset was used in the training and validation sets for the SASEN. The SASEN was trained on a desktop with an Intel Core i5-9600 CPU @ 3.70 GHz, 32 GB RAM, and NVidia Titan X Pascal.

To demonstrate the effectiveness of the proposed fault diagnosis strategy, two tests were performed. The first test was conducted on the experimental data while considering the transient state to be similar to that of the actual operating machine, as shown in Figure 6. The proposed method was tested by using data in which the load torque was transiently increased from 1 to 4 Nm for 18 s. The second test was performed on untrained data with various load torques. To achieve high accuracy in actual applications, it is essential to show that fault diagnoses can be performed under various operating conditions with an appropriate amount of training data. The proposed method was tested under untrained load torque conditions from 1.2 to 3.8 Nm in increments of 0.2 Nm.

### 4.3. Results and Discussion

#### 4.3.1. Test Results for Transient Load Torque

Figure 7 shows the test results for the ISCF cases with transient changes in the load torque. The fault diagnosis results for ISCF Cases 1, 2, and 3 are presented, and the test results for the healthy machine are also included for comparison. The SASEN outputs two fault indicators: ${\mathrm{FI}}_{\mathrm{ISCF}}$, which represents the severity of the ISCF (see Figure 7a), and ${\mathrm{FI}}_{\mathrm{DF}}$, which represents the severity of the DF (see Figure 7b). For the healthy IPMSM, the estimated ${\mathrm{FI}}_{\mathrm{ISCF}}$ and ${\mathrm{FI}}_{\mathrm{DF}}$ are 0.05 and 1.005, which are close to 0 and 1, respectively, even when the load torque changes transiently. The estimated fault indicators are similar to the actual values, suggesting that no fault has occurred. However, when the ISCF occurs, ${\mathrm{FI}}_{\mathrm{ISCF}}$ increases, but ${\mathrm{FI}}_{\mathrm{DF}}$ does not change as the shorted turn ratio μ increases from 0.042 to 0.125. This suggests that the proposed strategy can diagnose ISCFs. In addition, it can be seen that the ISCF is diagnosed by estimating the severity of the fault, even with the transient load torque. This is because the NSC and NSV reflect the severity of the ISCF, whereas the PSC reflects the change in the load torque and compensates for the change in the operating point to the network. In addition, the fault indicator for the DF does not change because the PSV, which reflects the demagnetization severity, does not change significantly with the ISCF. Therefore, the proposed strategy can diagnose an ISCF by estimating the severity, and can clearly distinguish between machines with the ISCFs and healthy machines, and also from a DF.

Figure 8 shows the test results for the DF cases with transient changes in the load torque. The fault diagnosis results for DFs with two severity levels are presented. Contrary to the ISCF diagnosis results, the estimated ${\mathrm{FI}}_{\mathrm{DF}}$ is reduced compared to that of the healthy machine, but the estimated ${\mathrm{FI}}_{\mathrm{ISCF}}$ is close to 0, similar to that of the healthy machine when the DF occurs. In addition, as the severity of DF increases from 83.3% to 66.7%, the estimated ${\mathrm{FI}}_{\mathrm{DF}}$ is further reduced and becomes closer to the actual value, suggesting that the proposed strategy can diagnose DFs. Meanwhile, the severity of DF is accurately estimated even with transient load torque, and it is possible to accurately distinguish DF from ISCF because ${\mathrm{FI}}_{\mathrm{ISCF}}$ does not respond to DF. This is because the PSV and PSC accurately reflect the decrease in flux, whereas the NSC does not increase with DF. Therefore, the results validate that the proposed strategy can diagnose DF by estimating its severity.

Figure 9 shows the test results for the HF cases with transient changes in the load torque. The fault diagnosis results for HF with six cases are presented. Both ${\mathrm{FI}}_{\mathrm{ISCF}}$ and ${\mathrm{FI}}_{\mathrm{DF}}$ are changed compared to the healthy machine, as the HF is the fault where the ISCF and DF occur simultaneously. The estimated ${\mathrm{FI}}_{\mathrm{ISCF}}$ increases as μ increases, and the estimated ${\mathrm{FI}}_{\mathrm{DF}}$ decreases as the IPMSM becomes more demagnetized. Remarkably, the HF has the same fault conditions as in the ISCF, but ${\mathrm{FI}}_{\mathrm{ISCF}}$ shows a reduced value compared to that with the ISCF. This is because the flux linkage from the PM to the stator winding is reduced owing to the demagnetization, which in turn reduces the magnitude of the fault current. Nevertheless, as the input signals reflect the characteristics of the faults, the SASEN can accurately estimate the severity of the faults. Therefore, it is demonstrated through experiments that the proposed strategy can diagnose HFs by estimating these two fault indicators.

#### 4.3.2. Test Results for Untrained Load Torque

Figure 10 shows the test results for the HF5 at untrained load torques from 1.2 to 3.8 Nm. For each load torque, three seconds of measurement data were used for the fault diagnosis test. The proposed strategy can diagnose the fault even for untrained load torques, even though the SASEN is trained only on the data of the four load torques from 1 to 4 Nm. The results show more errors and fluctuations compared with the results from Test 1. Nevertheless, the proposed strategy provides a fault severity estimation for the HF, and a clear distinction from a healthy IPMSM. Because the PSC reflects changes in the load torque and the NSC, NSV, and PSV reflect the severities of the ISCF and DF to the network, fault diagnosis is possible even with untrained data.

### 4.4. Comparison with Other Methods

To verify the superiority of the proposed strategy, the strategy was compared with the latest fault diagnosis methods via experiments. The fault diagnosis results are listed in Table 3. Untrained data were used to test each fault diagnosis method. The evaluation metrics were the root mean squared error (RMSE) for each fault indicator, and the time consumption for the test. The TCN [24] represents the latest study on fault detection and classification; it was used in a comparative study, because fault severity estimation studies are extremely rare, and because it shows good results in detection and classification. The activation and softmax functions were removed from the TCN for regression. The attention RNN [25] was also compared. The hyperparameters used for the TCN and attention RNN were those presented in each study, respectively, and inputs were the same as those for the proposed strategy, i.e., the NSV, NSC, PSV, and PSC. As can be seen from the results, the RMSE of ${\mathrm{FI}}_{\mathrm{ISCF}}$ shows the best result with the smallest value of 0.0566 in the proposed method. The attention RNN shows the smallest RMSE for ${\mathrm{FI}}_{\mathrm{ISCF}}$, but there is no significant difference relative to the proposed method. Moreover, the time consumption for testing is 516.4 ms with the attention RNN, i.e., much larger than the 40.6 ms required for the proposed method. This is because the attention RNN requires sequential operations, thereby increasing the computational complexity. In contrast, the proposed strategy enables parallel operations without the sequential operations used in the RNN, resulting in a much shorter test time. Similar to the proposed strategy, the TCN can be tested quickly with parallel operations, but the accuracy of estimating the fault severity is much lower than that in the proposed strategy; the ${\mathrm{FI}}_{\mathrm{ISCF}}$ and ${\mathrm{FI}}_{\mathrm{DF}}$ show four and seven times greater RMSEs, respectively. This is because in the proposed strategy, the decoder and encoder–decoder attention used behind the encoder further improve the regression performance. Therefore, the proposed strategy exhibits the best fault diagnosis performance for severity estimations.

Table 4 presents qualitative comparisons with other fault diagnosis methods to further evaluate the advances of the proposed method. Model-based methods [17,18] can diagnose only one fault by estimating the severity and require accurate models. Data-driven methods [20,21,23,25] do not require accurate models, but most studies do not estimate the severity of faults. The proposed method diagnoses the ISCF, DF, and HF by estimating fault severities and does not require an accurate model. In addition, the proposed method demonstrates that generalization capability as fault diagnosis is possible even under varying load torque and untrained load torque.

## 5. Conclusions

In this study, we proposed a novel fault diagnosis strategy for ISCFs, DFs, and HFs in a PMSM by using the SASEN with an encoder–decoder structure based on a self-attention module. The SASEN estimates fault indicators for quantifying the severity of each fault, and uses the PSV, PSC, NSV, and NSC as inputs. An MSA approach is used for the encoder and decoder to improve the regression ability of the network, i.e., by focusing on a specific feature representation. The proposed method does not require accurate models and parameters to estimate the severity of faults, and is believed to represent the first study to diagnose an HF by estimating its severity. Experiments implementing the ISCF, DF, and HF validate that the proposed strategy successfully estimates the fault indicators and diagnoses the faults under various fault and load torque conditions. In a future study, we will focus on the severity estimations for the various faults occurring in a PMSM with varying speeds and untrained operating conditions.

## Figures and Tables

**Figure 1 sensors-22-04639-f001:**
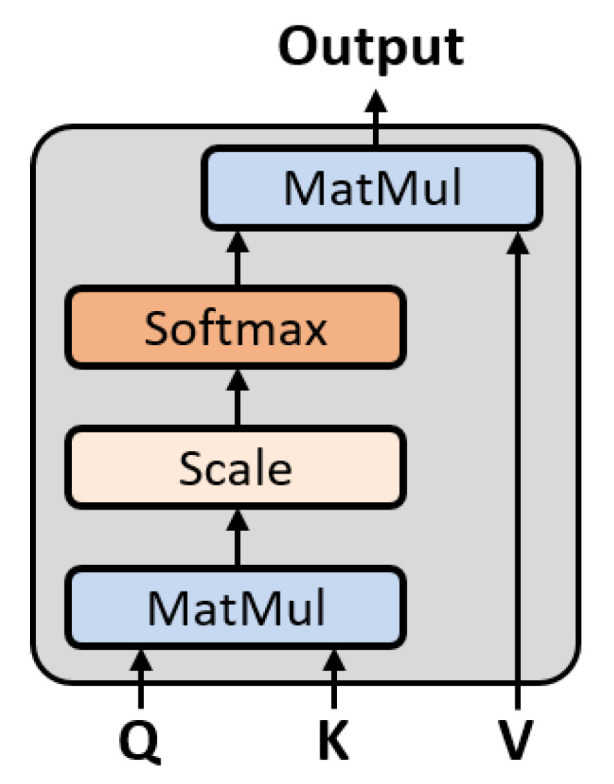
Structure of self-attention module.

**Figure 2 sensors-22-04639-f002:**
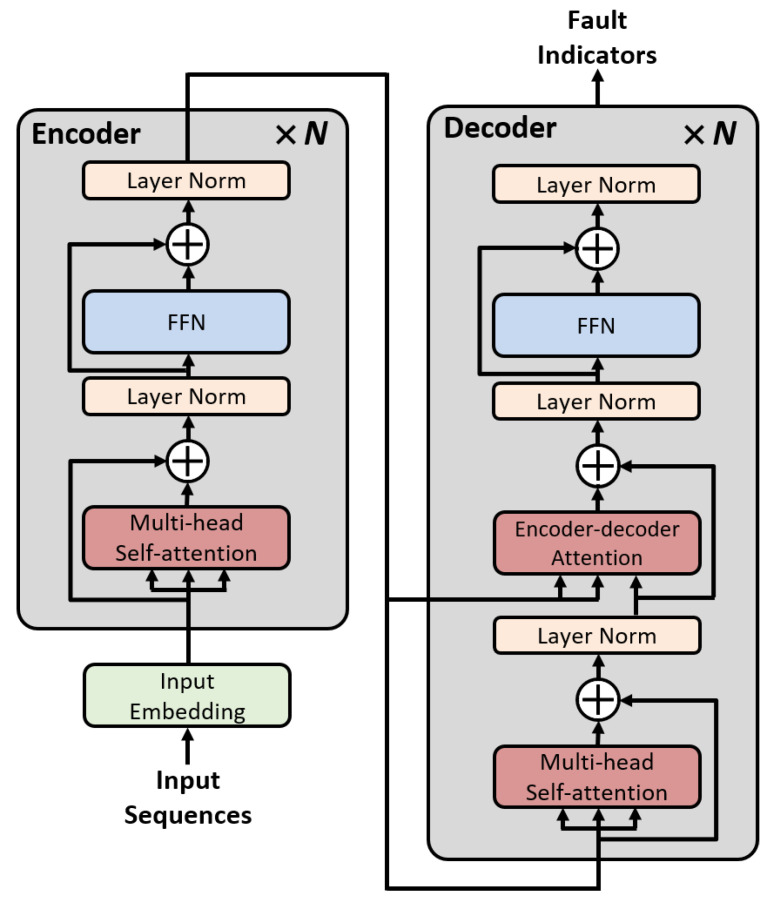
Overall architecture of the self-attention-based severity estimation network (SASEN).

**Figure 3 sensors-22-04639-f003:**
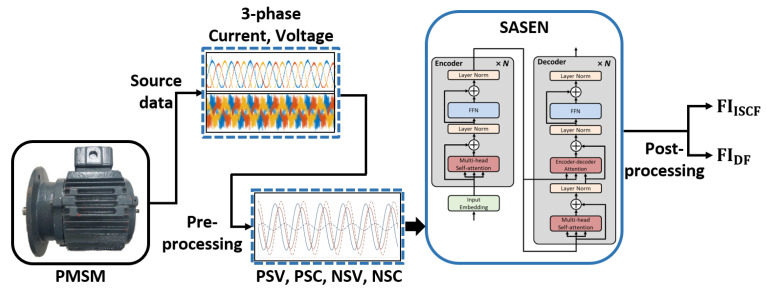
Overall structure of fault diagnosis system.

**Figure 4 sensors-22-04639-f004:**
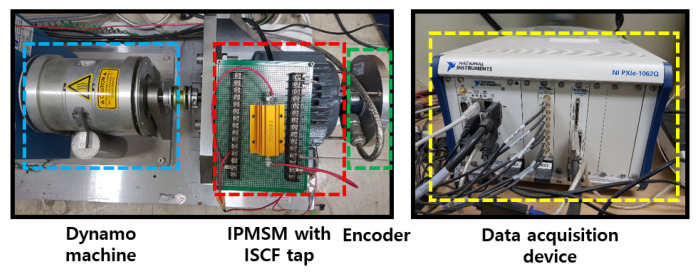
Experimental setup of the interior permanent-magnet synchronous machine (IPMSM).

**Figure 5 sensors-22-04639-f005:**
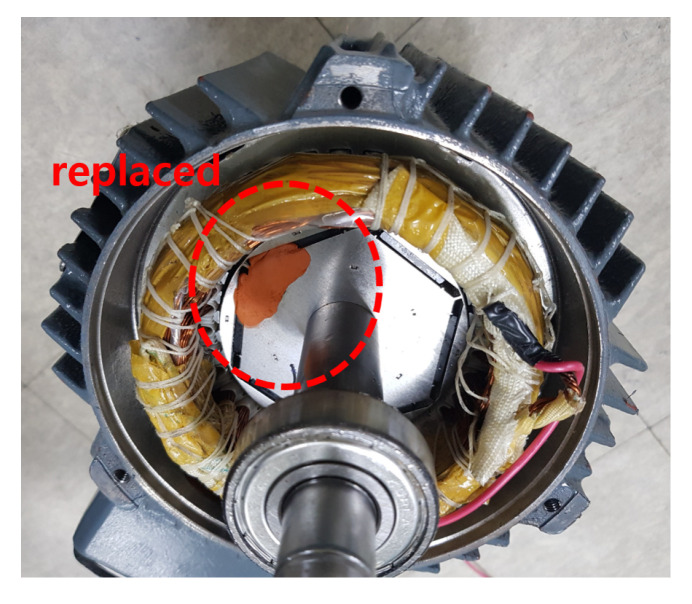
Demagnetized IPMSM. Permanent magnets are replaced with dummies.

**Figure 6 sensors-22-04639-f006:**
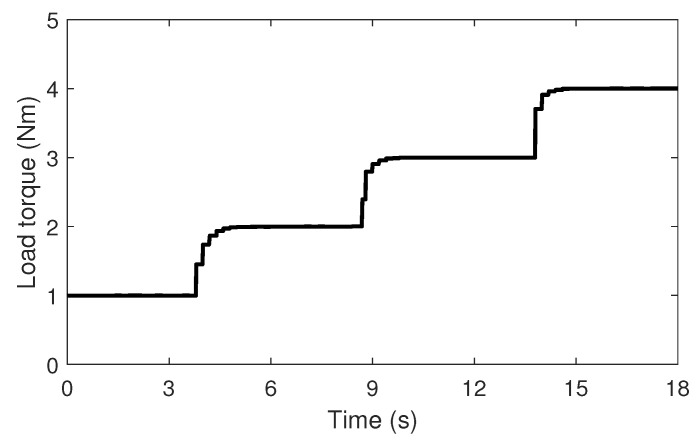
Change in the load torque for the test 1.

**Figure 7 sensors-22-04639-f007:**
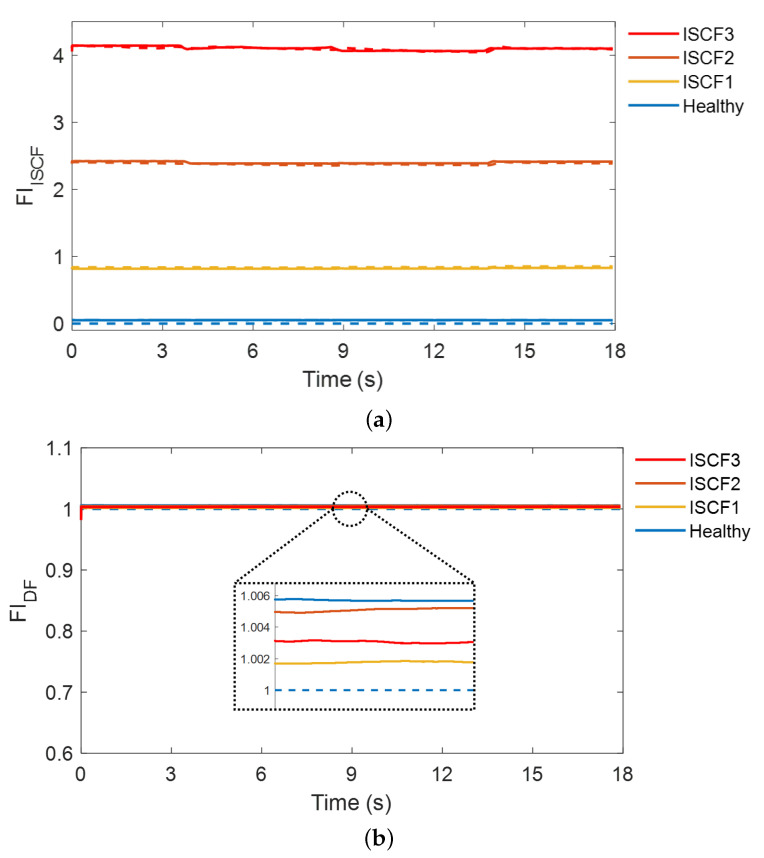
Test results for the interturn short-circuit fault (ISCF) at transient load torque. (**a**) Fault indicator for the ISCF. (**b**) Fault indicator for the demagnetization fault (DF). The solid line and dashed line represent the estimated and real fault indicators, respectively.

**Figure 8 sensors-22-04639-f008:**
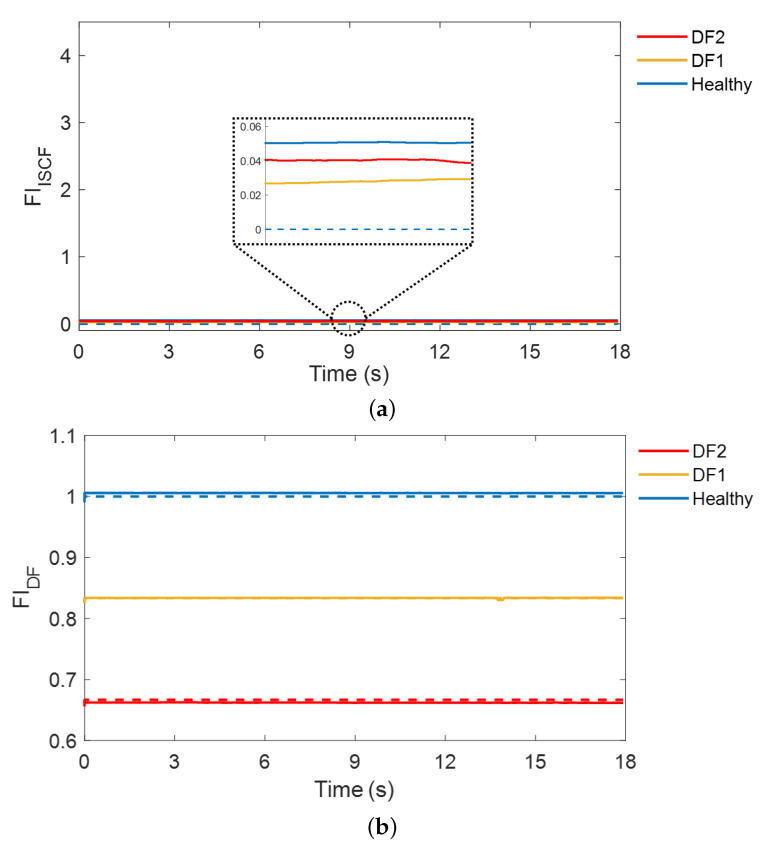
Test results for the DF at transient load torque. (**a**) Fault indicator for the ISCF. (**b**) Fault indicator for the DF. The solid line and dashed line represent the estimated and real fault indicators, respectively.

**Figure 9 sensors-22-04639-f009:**
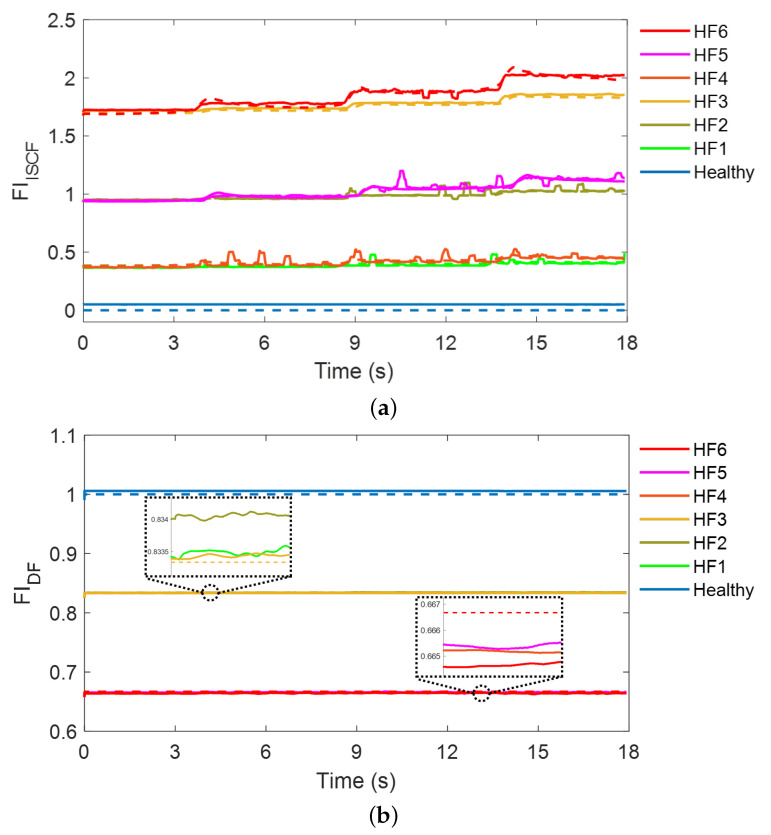
Test results for the hybrid fault (HF) at transient load torque. (**a**) Fault indicator for the ISCF. (**b**) Fault indicator for the DF. The solid line and dashed line represent the estimated and real fault indicators, respectively.

**Figure 10 sensors-22-04639-f010:**
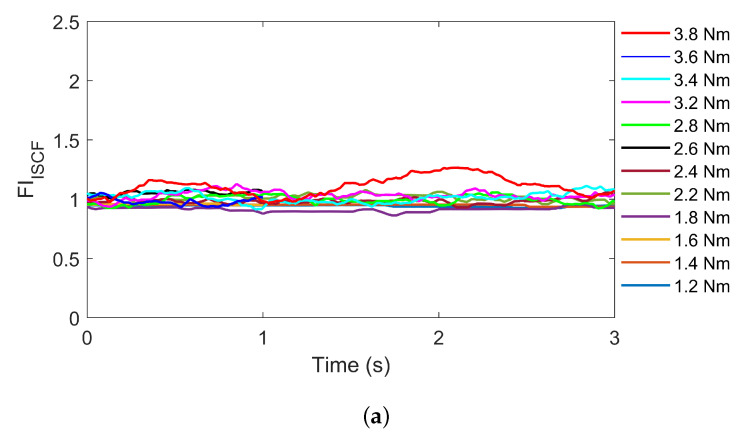
Test results for the untrained load torque. Test case is HF5 at untrained load torques of 1.2 to 3.8 Nm. (**a**) Fault indicator for the ISCF. (**b**) Fault indicator for the DF.

**Table 1 sensors-22-04639-t001:** Detailed specifications of the interior permanent-magnet synchronous machine (IPMSM).

Parameters	Values
Power	2.2 kW
Rated torque	4.7 Nm
Rated speed	4500 rpm
Rated current	8.4 A
*d*-axis inductance	2.836 mH
*q*-axis inductance	5.999 mH
Stator resistance	0.43 Ω
Back EMF constant	36 V/kr/min

**Table 2 sensors-22-04639-t002:** Description of fault cases for interturn short-circuit faults (ISCFs), demagnetization faults (DFs), and hybrid faults (HFs), and operating conditions.

Case	Healthy	ISCF1	ISCF2	ISCF3	DF1	DF2
$r_{f}$ (Ω)	inf	0.218	0.218	0.218	inf	inf
μ	0	0.042	0.083	0.125	0	0
$\psi_{m,f}/\psi_{m,h}$	1	1	1	1	0.833	0.667
**Case**	**HF1**	**HF2**	**HF3**	**HF4**	**HF5**	**HF6**
$r_{f}$ (Ω)	0.217	0.217	0.217	0.217	0.217	0.217
μ	0.042	0.083	0.125	0.042	0.083	0.125
$\psi_{m,f}/\psi_{m,h}$	0.833	0.833	0.833	0.667	0.667	0.667
Speed (rpm)	4500					
Torque (Nm)	1, 2, 3, 4					

**Table 3 sensors-22-04639-t003:** Comparison of diagnostic results including root mean squared error (RMSE) for the untrained load torque for the proposed method, transformer convolution network (TCN), and attention recurrent neural network (RNN).

	RMSE of ${\mathrm{FI}}_{\mathrm{ISCF}}$	RMSE of ${\mathrm{FI}}_{\mathrm{DF}}$	Test Time (ms)
TCN [24]	0.225	0.0126	28.8
Attention RNN [25]	0.0879	0.0006	516.4
**Proposed**	0.0566	0.0019	40.6

**Table 4 sensors-22-04639-t004:** Comparison with other fault diagnosis methods.

	Proposed Method	[17]	[18]	[20]	[21]	[23]	[25]
Fault diagnosis for ISCF	**O**	**O**	X	X	X	**O**	**O**
Fault diagnosis for DF	**O**	X	**O**	**O**	**O**	X	X
Fault diagnosis for HF	**O**	X	X	X	**O**	X	X
Estimation of fault severity	**O**	**O**	**O**	X	X	X	**O**
No need for accurate model	**O**	X	X	**O**	**O**	**O**	**O**
Fault diagnosis under varying torque	**O**	**O**	**O**	**O**	**O**	X	**O**

The bold meaning is to highlight better indices. **O** is better.

## Data Availability

The data presented in this study are available on request from the corresponding author. The data are not publicly available because it is company confidential information.

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
