# Peer review of "Severity Estimation for Interturn Short-Circuit and Demagnetization Faults through Self-Attention Network"

_sensors, 2022, doi:10.3390/s22124639_

Round 1
Reviewer 1 Report
Nice work and nice paper. I've minor comments.
There are few places where I would suggest checking the standard English spellings. For example, preprocessed --> pre-processed. There are few other instances.
line 242: 20 and 35 A---> 20 A and 35 A
Reviewer 2 Report
In this submitted manuscript, a self-attention-based SEN is proposed for ISCF and DF fault diagnosis. The effectiveness of the method is experimentally validated. The fault diagnosis can be useful and important for PMSMs and therefore the topic can be useful to this field. I have the following comments for the authors. Hopefully, this can help further improve the quality of the work.
1. The authors claim that the proposed method can diagnose HF, i.e. when ISCF and DF occur simultaneously. Usually ISCF will occur first and then the DF. Therefore I assume capturing ISCF in the early stage is better than detecting ISCF and DF occurring simultaneously. So the question is why simultaneous capture is important?
2. Rather than comparing with conventional methods, the authors should highlight the advantages over existing data-driven methods, it is better to give a table to show what are the advances this proposed method can achieve, while other existing data-driven methods cannot.
3. Can the authors comment on the convergence capability of this method. In the paper, the residual connection is used to avoid gradient vanishing. Can the authors show some results with and without such treatment?
4. Can the authors comment on the impact of the network size on the accuracy of the method.
5. Can the authors comment on the sensitivity of this proposed method, i.e. to what level the fault starts to occur, the proposed method can successfully detect.
Reviewer 3 Report
The paper entitled “Severity Estimation for Interturn Short-Circuit and Demagnetization Faults Through Self-Attention Network” presents a method for the detection and severity estimation of interturn short-circuit faults and demagnetization faults.
Although, very good literature review has been done by the authors, some previous literature than the ones cited, and some important authors have been left aside. For example, a seminal paper analyzing this same topic with convolutional neural networks is:
Skowron M, Orlowska-Kowalska T, Wolkiewicz M, Kowalski CT (2020) Convolutional neural network-based stator current data-driven incipient stator fault diagnosis of inverter-fed induction motor. Energies 13:. https://doi.org/10.3390/en13061475
And from the same authors:
Skowron M, Wolkiewicz M, Tarchała G (2020) Stator winding fault diagnosis of induction motor operating under the field-oriented control with convolutional neural networks. Bull Polish Acad Sci Tech Sci 68:1039–1048. https://doi.org/10.24425/bpasts.2020.134660
Skowron M, Orlowska-Kowalska T, Kowalski CT (2022) Detection of permanent magnet damage of PMSM drive based on direct analysis of the stator phase currents using convolutional neural network. IEEE Trans Ind Electron 0046:1–1. https://doi.org/10.1109/tie.2022.3146557
Also, some other relevant literature on the demagnetization topic is missing:
R. E. Quintal-Palomo, M. Flota-Bañuelos, A. Bassam, R. Peón-Escalante, F. Peñuñuri and M. Dybkowski, "Post-Fault Demagnetization of a PMSG Under Field Oriented Control Operation," in IEEE Access, vol. 9, pp. 53838-53848, 2021, doi: 10.1109/ACCESS.2021.3070531
Some explanation of the training done on the neural network is missing. It is stated that a GPU was used in the PC although no explanation on how this impacts training or if some of the code was running in parallel.
Some other minor corrections should be done, i.e., no explanation of the abbreviation EF (it is believed to stand for Eccentricity Faults)
